# Dirichlet Polynomials and Entropy

**DOI:** 10.3390/e23081085

**Published:** 2021-08-21

**Authors:** David I. Spivak, Timothy Hosgood

**Affiliations:** Topos Institute, Berkeley, CA 94704, USA; david@topos.institute

**Keywords:** bundle, weighted geometric mean, category theory, Dirichlet polynomial

## Abstract

A Dirichlet polynomial *d* in one variable y is a function of the form $d\left(y\right)=a_{n}n^{y}+\cdots +a_{2}2^{y}+a_{1}1^{y}+a_{0}0^{y}$ for some $n,a_{0},\ldots ,a_{n}\in N$. We will show how to think of a Dirichlet polynomial as a set-theoretic bundle, and thus as an empirical distribution. We can then consider the Shannon entropy H(d) of the corresponding probability distribution, and we define its *length* (or, classically, its *perplexity*) by $L\left(d\right)=2^{H(d)}$. On the other hand, we will define a rig homomorphism h:Dir→Rect from the rig of Dirichlet polynomials to the so-called *rectangle rig*, whose underlying set is $R_{⩾0}\times R_{⩾0}$ and whose additive structure involves the weighted geometric mean; we write h(d)=(A(d),W(d)), and call the two components *area* and *width* (respectively). The main result of this paper is the following: the rectangle-area formula A(d)=L(d)W(d) holds for any Dirichlet polynomial *d*. In other words, the entropy of an empirical distribution can be calculated entirely in terms of the homomorphism *h* applied to its corresponding Dirichlet polynomial. We also show that similar results hold for the cross entropy.

## 1. Introduction

The purpose of this paper is simply to provide another categorical treatment of *entropy* of probability distributions, which turns out to be computed in terms of a rig homomorphism; our treatment also generalises to *cross entropy* (and thus to *Kullback–Leibler divergence*). What is particularly interesting about the treatment outlined here is that we can somewhat “visualise” the notion of entropy in terms of sizes of coding schemes (cf. Section 6). Not only that, but classical entropy is only homomorphic in the product of distributions, whereas the notion that we describe here is homomorphic in both the product and the sum.

A brief outline of this paper is as follows:Section 2: We recall the definitions of *Dirichlet polynomials* and *set-theoretic bundles*, along with their rig structures, from [1] (one important thing to note is the following: Dirichlet polynomials are well-studied objects in the setting of complex analysis, but we *cannot* apply tools from this area to our setting, because we have only *natural number* coefficients and *non-negative* exponents); we then study the equivalence between these two notions.Section 3: We explain how *empirical probability distributions* correspond to set-theoretic bundles (and thus to Dirichlet polynomials).Section 4: We define the *rig homomorphism* h:Dir→Rect that we wish to study, whose codomain is a rig encoding the weighted geometric mean; we prove some useful computational results and give some explicit examples.Section 5: We define the *entropy* H(d) of a Dirichlet polynomial using the classical notion of Shannon entropy; we give some explicit examples; we prove the main result of this paper (Theorem 1), relating entropy to the rig homomorphism defined in the previous section.Section 6: We try to provide some intuition for the image h(d) of a Dirichlet polynomial under the rig homomorphism, in terms of *coding schemes*.Section 7: We generalise Theorem 1 to the case of *cross-entropy*, or *Kullback–Leibler divergence*.

### Prerequisites

We assume that the reader is familiar with the fundamentals of category theory, such as functors, natural transformations, and (co)products, as covered, for example, in [2] (Chapter 1).

## 2. Dirichlet Polynomials and Bundles

This section is simply a brief summary of content from [1], repeated here for the convenience of the reader.

**Definition** **1.***A* Dirichlet polynomial *d in one variable y is a function of the form *
$$d\left(y\right)=a_{n}n^{y}+\ldots +a_{2}2^{y}+a_{1}1^{y}+a_{0}0^{y}$$
*for some $n,a_{0},\ldots ,a_{n}\in N$.**The set of Dirichlet polynomials is clearly closed under addition, and further under multiplication (using the distributive law along with the fact that $m^{y}\cdot n^{y}={\left(m\cdot n\right)}^{y}$). In fact, it has the structure of a *rig*: a “ring without negatives” (or, to be pedantic, a monoid object in commutative monoids). We denote this rig by Dir, where the additive unit is 0, and the multiplicative unit is $1^{y}$.*
*Note that we can embed N as a sub-rig of Dir, by $a\mapsto a\cdot 1^{y}$; we often use this fact and simply write a∈Dir.*


Following [1], we can think of Dirichlet polynomials as functors ${\mathrm{FSet}}^{op}\to \mathrm{FSet}$, where FSet is the category of finite sets. Indeed, given a natural number n∈N, the *exponential*
$n^{y}$ can be thought of as the Yoneda embedding of the set with *n* elements, i.e.,
$$n^{y}=\mathrm{FSet}\;\left(-,\underline{n}\right)$$
where $\underline{n}=\left\{1,\ldots ,n\right\}$. (For typographical convenience, we sometimes use the notation *n* and $\underline{n}$ interchangeably. In particular, we write e.g., d(0) instead of $d(\underline{0})$). Then the addition of exponentials corresponds to the coproduct of the corresponding representable functors (and so multiplication by a natural number $a_{n}$ corresponds to the $a_{n}$-fold coproduct of the representable functor with itself). This means that evaluating a Dirichlet polynomial at some natural number *n* corresponds to evaluating the corresponding functor on the finite set $\underline{n}$.

Note that $0^{y}$ is *not* the initial object 0, since
$$0^{\underline{n}}=\left\{\begin{array}{cc} 1 & \mathrm{if}\;n=0;\\ 0 & \mathrm{if}\;n⩾1 \end{array}\right.$$
i.e., $0^{y}\neq 0$.

**Example** **1.**
*The Dirichlet polynomial*
$$d\left(y\right)=4^{y}+4\cdot 1^{y}$$
*evaluated at *0* gives*
$$\begin{array}{cc} d(0) & =\mathrm{FSet}\;\left(\underline{0},\underline{4}\right)⊔\left({⊔}_{i=1}^{4}\mathrm{FSet}\;\left(\underline{0},\underline{1}\right)\right)\\ & ≅\underline{1}⊔\underline{4}\\ & ≅\underline{5}, \end{array}$$
*and, similarly,*
$$\begin{array}{cc} d(1) & =\mathrm{FSet}\;\left(\underline{1},\underline{4}\right)⊔\left({⊔}_{i=1}^{4}\mathrm{FSet}\;\left(\underline{1},\underline{1}\right)\right)\\ & ≅\underline{4}⊔\underline{4}\\ & ≅\underline{8}. \end{array}$$

*Note that, since $1^{y}=1$, we can write $d\left(y\right)=4^{y}+4$.*


**Definition** **2.***A* morphism *φ:d→e of Dirichlet polynomials is a natural transformation of (contravariant) functors. Denote by Dir the category of Dirichlet polynomials (thought of as functors ${\mathrm{FSet}}^{op}\to \mathrm{FSet}$), and by Dir(d,e) the set of all morphisms d→e.*

When we think of Dirichlet polynomials as functors ${\mathrm{FSet}}^{op}\to \mathrm{FSet}$, addition is given by the coproduct (disjoint union of sets), and multiplication by the product (Cartesian product of sets). This means that working with Dirichlet polynomials in Dir really is like working with polynomials, in the sense that addition and multiplication are exactly “as expected”.

**Example** **2.**
*The only slightly confusing aspect of multiplication in Dir is how $0^{y}$ behaves (since $0^{y}\neq 0$): if d(y) is a Dirichlet polynomial, then*
$$d\left(y\right)\cdot 0^{y}=d\left(0\right)\cdot 0^{y},$$
*as follows from the aforementioned fact that $0^{\underline{n}}$ is zero for n≠0, and *1* for n=0.*

*We can use this general fact for specific computations. For example, let*
$$\begin{array}{cc} d(y) & :=3\cdot 2^{y}+1^{y}\\ e(y) & :=4^{y}+2^{y}+3\cdot 0^{y} \end{array}$$

*Then*
$$\begin{array}{cc} \left(d\cdot e)(y\right) & =\left(3\cdot 8^{y}+3\cdot 4^{y}+9\cdot 0^{y}\right)+\left(4^{y}+2^{y}+3\cdot 0^{y}\right)\\ & =3\cdot 8^{y}+4\cdot 4^{y}+2^{y}+12\cdot 0^{y}. \end{array}$$
*(and $d+e=4^{y}+4\cdot 2^{y}+1^{y}+3\cdot 0^{y}$).*


There is a more geometric interpretation of objects of Dir as *set-theoretic bundles*, i.e., (isomorphism classes of) functions E→B, where E,B∈FSet, given as follows: to the Dirichlet polynomial $d:{\mathrm{FSet}}^{op}\to \mathrm{FSet}$, we associate the function $\pi_{d}=d\left(!\right):d\left(1\right)\to d\left(0\right)$ induced by the unique function $!:\underline{0}\to \underline{1}$. (This vague statement can be upgraded to an equivalence of categories: ([1], [Theorem 4.6]) for more details).

For example, to the polynomial $d\left(y\right)=4^{y}+4\cdot 1^{y}$, we associate the bundle



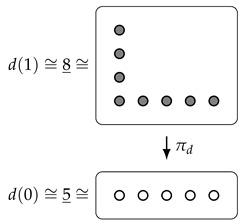



Note that bundles also form a rig, where the sum is given by disjoint union of sets, and the product is given by the Cartesian product of sets, both on the base and the total space. Further, the equivalence between Dirichlet polynomials and bundles respects the rig structures (cf. [1], Theorem 4.6). Because of this, we often switch freely between thinking of Dirichlet polynomials as functors $d:{\mathrm{FSet}}^{op}\to \mathrm{FSet}$, as bundles $\pi_{d}:d\left(1\right)\to d\left(0\right)$, and simply as functions of the form $\sum_{j=0}^{n}a_{j}\cdot j^{y}$.

**Example** **3.**
*We can draw the bundle corresponding to $\left(2^{y}+1\right)\cdot \left(2^{y}+1\right)$ as follows:*


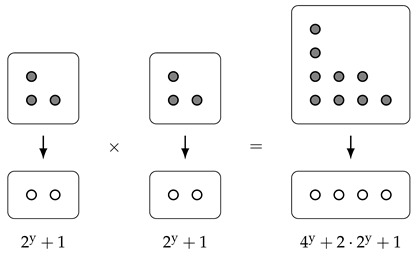



**Lemma** **1.**
*Let $d\left(y\right)\sum_{j=0}^{n}a_{j}\cdot j^{y}$ be a Dirichlet polynomial. Then*
$$\begin{array}{cc} \left|d(0)\right| & =\sum_{j=0}^{n}a_{j}\\ \left|d(1)\right| & =\sum_{j=0}^{n}a_{j}j. \end{array}$$


**Proof.** This follows from the fact that $n^{\underline{0}}=1$ and $n^{\underline{1}}=n$, for all n∈N. □

**Definition** **3.**
*Let d∈Dir. For i∈d(0), we define*
$$d\left[i\right]\pi_{d}^{-1}\left(i\right)$$
*where $\pi_{d}:d\left(1\right)\to d\left(0\right)$ is the bundle corresponding to d.*


Using the fact that the sum of bundles is given by the disjoint union of sets, we can use this above definition to write any Dirichlet polynomial *d* as
$$d\left(y\right)≅\sum_{i\in d(0)}d{\left[i\right]}^{y}$$
(where ∑ is the coproduct in FSet).

**Corollary** **1.**
*Let $d(y):=\sum_{i\in d(0)}d{\left[i\right]}^{y}$ be a Dirichlet polynomial. Then*
$$\left|d\left(1\right)\right|=\sum_{i\in d(0)}\left|d\left[i\right]\right|.$$


**Proof.** This is, again, simply the fact that $n^{\underline{1}}=n$ for all n∈N. □

**Lemma** **2.**
*A morphism φ:d→e of Dirichlet polynomials is exactly a morphism of the corresponding bundles, i.e., functions $\varphi_{0}:d\left(0\right)\to e\left(0\right)$ and $\varphi_{1}:d\left(1\right)\to e\left(1\right)$ such that*


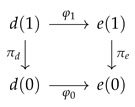

*commutes.*


**Proof.** This statement forms a specific part of ([1], Theorem 4.6), but the proof is simple enough that we give a direct version here. Writing $d\left(y\right)\sum_{i\in d(0)}d{\left[i\right]}^{y}$ and $e\left(y\right)\sum_{i\in e(0)}e{\left[i\right]}^{y}$, we see that
$$\begin{array}{cc} {\mathrm{Hom}}_{\left[{\mathrm{FSet}}^{op},\mathrm{FSet}\right]}\left(d,e\right) & ≅\prod_{i\in d(0)}{\mathrm{Hom}}_{\left[{\mathrm{FSet}}^{op},\mathrm{FSet}\right]}\left(d{\left[i\right]}^{y},\sum_{j\in e(0)}e{\left[j\right]}^{y}\right)\\ & ≅\prod_{i\in d(0)}\sum_{j\in e(0)}e{\left[j\right]}^{d[i]}\\ & =\prod_{i\in d(0)}\sum_{j\in e(0)}{\mathrm{Hom}}_{\mathrm{FSet}}\left(d\left[i\right],e\left[j\right]\right) \end{array}$$
(the first isomorphism is by the universal property of the coproduct; the second isomorphisms is the Yoneda lemma). However, an element of this set is exactly a bundle morphism: we have, for all i∈d(0), some j∈e(0) along with a function d[i]→e[j]; $\varphi_{1}$ is given by the disjoint union of all these d[i]→e[j], and $\varphi_{0}$ is given by the the choice of *j* for each *i*. □

**Definition** **4.**
*Given Dirichlet polynomials d,e∈Dir such that d(0)=e(0), we denote by ${\mathrm{Dir}}_{/d(0)}\left(d,e\right)$ the set of morphisms $(\varphi_{0},\varphi_{1}):d\to e$ such that $\varphi_{0}=id$.*


Given the correspondence between Dirichlet polynomials and bundles, we might rightly ask why we should prefer to work with the former over the latter. For one possible answer to this, see Section 6.

## 3. Bundles as Empirical Distributions

The interpretation of Dirichlet polynomials as bundles helps us to understand how they relate to probability theory. Imagine flipping a coin eight times and observing five heads and three tails; we refer to “heads” and “tails” as *outcomes*, and each of the eight flips as *draws*; every draw has an associated outcome.

Consider some bundle $\pi_{d}:d\left(1\right)\to d\left(0\right)$. We can think of d(0) as the set of outcomes, and d(1) as the set of draws; the fibre $\pi_{d}^{-1}\left(x\right)$ over an outcome x∈d(0) corresponds to all the draws that lead to the outcome *x*, and so we obtain a probability distribution on d(0) by setting $P\left(X=x\right)=\frac{|\pi_{d}^{-1}\left(x\right)|}{\left|d(0)\right|}$. Conversely, any *rational distribution* (i.e., a distribution such that all probabilities are rational numbers ) on a finite set arises in this way: take the finite set as the set of outcomes; take the least common multiple of the denominators of all the probabilities as the cardinality of d(1); and then take P(X=x)·|d(1)| many elements of d(1) to be in the fibre of x∈d(0).

**Example** **4.**
*Consider the set $S=\{x_{1},x_{2},x_{3},x_{4}\}$, endowed with the probability distribution such that*
$$\begin{array}{cc} P(x_{1}) & =\frac{1}{5}\;P\left(x_{2}\right)=\frac{1}{6}\\ P(x_{3}) & =\frac{1}{2}\;P\left(x_{4}\right)=\frac{2}{15} \end{array}$$

*Define the sets $d\left(0\right)=\underline{4}$ and $d\left(1\right)=\underline{30}$, and define the function $\pi_{d}:d\left(1\right)\to d\left(0\right)$ by*
$$\pi \left(n\right)=\left\{\begin{array}{cc} 1 & if\;0⩽n<6;\\ 2 & if\;6⩽n<11;\\ 3 & if\;11⩽n<26;\\ 4 & if\;26⩽n<30. \end{array}\right.$$

*Then the empirical probability distribution on the bundle π:d(1)→d(0) agrees exactly with the given distribution on S. As a Dirichlet polynomial, this bundle is given (up to relabelling the outcomes) by $d(y):=15^{y}+6^{y}+5^{y}+4^{y}$.*

*Note that any multiple $m^{y}\cdot d\left(y\right)$ of d (for m⩾1) will correspond to the same probability distribution as d itself, but to a different empirical distribution, since it will have m times as many draws.*


Under this interpretation of Dirichlet polynomials as empirical distributions, multiplication d·e corresponds to taking the *product distribution*.

**Remark** **1.**
*For any d∈Dir, and any n∈N, we can give |d(n)| a combinatorial interpretation: it is the number of ways of choosing n indistinguishable (in the sense that they have the same outcome) draws, i.e., the number of length-n lists of elements of d[i] for some i∈d(0).*

*To see this, note that $d\left(n\right)=\mathrm{Dir}\;\left(n^{y},d\right)$ (by Yoneda), and so d(n) is in bijection with the set of bundle morphisms $\left(\varphi_{0},\varphi_{1}\right):\left(n\to 1\right)\to \left(d\left(1\right)\to d\left(0\right)\right)$, which are given exactly by choosing n (possibly repeated) elements of d(1) that all lie in the same fibre (namely the fibre above the point specified by $\varphi_{0}\left(1\right)$).*


**Remark** **2.***Although we deal only with* finite *sets and* rational *probability distributions here, it seems likely that one could follow the methods of [3] and consider colimits of these to obtain analogous results for* arbitrary *probability distributions on* discrete measurable spaces.

## 4. Area and Width

**Definition** **5.**
*Define the rig Rect as follows. The underlying set is $R_{⩾0}\times R_{⩾0}$. The multiplicative structure has unit (1,1), and is given by component-wise multiplication:*
$$\left(A_{1},W_{1}\right)\cdot (A_{2},W_{2}):=\left(A_{1}A_{2},W_{1}W_{2}\right).$$

*The additive structure has unit (0,0), and is given by real-number addition in the first component, and by weighted geometric mean in the second component:*
$$\left(A_{1},W_{1}\right)+(A_{2},W_{2}):=\left(A_{1}+A_{2},{\left(W_{1}^{A_{1}}W_{2}^{A_{2}}\right)}^{\frac{1}{A_{1}+A_{2}}}\right).$$
*Given an element (A,W) in Rect, we call A its *area *and W its* width.

The fact that Rect is indeed a rig follows from the fact that its multiplication distributes over its addition:
$$\begin{array}{cc} \left(A_{1},W_{1}\right)\cdot \left(\left(A_{2},W_{2}\right)+\left(A_{3},W_{3}\right)\right) & =\left(A_{1},W_{1}\right)\cdot \left(A_{2}+A_{3},{\left(W_{2}^{A_{2}}W_{3}^{A_{3}}\right)}^{\frac{1}{A_{2}+A_{3}}}\right)\\ & =\left(A_{1}\left(A_{2}+A_{3}\right),W_{1}{\left(W_{2}^{A_{2}}W_{3}^{A_{3}}\right)}^{\frac{1}{A_{2}+A_{3}}}\right)\\ & =\left(A_{1}A_{2}+A_{1}A_{3},{\left(W_{1}^{A_{2}+A_{3}}W_{2}^{A_{2}}W_{3}^{A_{3}}\right)}^{\frac{1}{A_{2}+A_{3}}}\right)\\ & =\left(A_{1}A_{2}+A_{1}A_{3},{\left({\left(W_{1}W_{2}\right)}^{A_{2}}{\left(W_{1}W_{3}\right)}^{A_{3}}\right)}^{\frac{1}{A_{2}+A_{3}}}\right)\\ & =\left(A_{1}A_{2}+A_{1}A_{3},{\left({\left(W_{1}W_{2}\right)}^{A_{1}A_{2}}{\left(W_{1}W_{3}\right)}^{A_{1}A_{3}}\right)}^{\frac{1}{A_{1}A_{2}+A_{1}A_{3}}}\right)\\ & =\left(A_{1}A_{2},W_{1}W_{2}\right)+\left(A_{1}A_{3},W_{1}W_{3}\right)\\ & =\left(A_{1},W_{1}\right)\cdot \left(A_{2},W_{2}\right)+\left(A_{1},W_{1}\right)\cdot \left(A_{3},W_{3}\right). \end{array}$$

**Proposition** **1.**
*There exists a unique rig morphism h:Dir→Rect for which*
$$h:n^{y}\mapsto \left(n,n\right).$$


**Proof.** Since every Dirichlet polynomial is just a sum of exponentials, a rig homomorphism is fully determined by its action on exponentials, since it must respect addition. So we just need to show that *h* does indeed extend to a rig homomorphism, but this follows from the fact that $m^{y}\cdot n^{y}={\left(m\cdot n\right)}^{y}$. □

**Definition** **6.***Given a Dirichlet polynomial d, we define its *area *A(d) and its* width *W(d) to be given by the components of h(d)=(A(d),W(d)).*

With this definition, along with Proposition 1, we see that
$$A\left(n^{y}\right)=W\left(n^{y}\right)=n.$$

**Lemma** **3.**
*Let d∈Dir and a∈N. Then*
*1.* 
*h(a)=(a,1);*
*2.* 
*A(a·d)=aA(d);*
*3.* 
*W(a·d)=W(d).*



**Proof.** Recall that addition (and thus scalar multiplication) in Rect involves the weighted geometric mean in the second component. Then
$$\begin{array}{cc} h(a) & :=h(a\cdot 1^{y})\\ & =a\cdot h(1^{y})\\ & =a\cdot (1,1)\\ & =(a,1) \end{array}$$
which proves (i). For (ii) and (iii), since *h* is a rig homomorphism (and thus respects addition), it suffices to consider the case where *d* is an exponential, say $d\left(y\right)=n^{y}$. However, then
$$\begin{array}{cc} h(a\cdot d) & =a\cdot h(d)\\ & =a\cdot (n,n)\\ & =\left(an,{\left(\underset{atimes}{\underbrace{n^{n}n^{n}\ldots n^{n}}}\right)}^{1/an}\right)\\ & =\left(an,{\left(n^{an}\right)}^{1/an}\right)\\ & =(an,n) \end{array}$$
i.e., A(a·d)=aA(d) and W(a·d)=W(d), as claimed. □

**Corollary** **2.**
*Let d∈Dir. Then*
$$\begin{array}{cc} A(d) & =|d(1)|\\ W{\left(d\right)}^{A(d)} & =\left|{\mathrm{Dir}}_{/d(0)}\left(d,d\right)\right|. \end{array}$$


**Proof.** Write $d\left(y\right)=a_{n}\cdot n^{y}+\ldots +a_{1}\cdot 1^{y}+a_{0}\cdot 0^{y}$. Using Lemma 3, along with Definition 5, we see that
$$\begin{array}{cc} h(d) & =\left(a_{n}n,n\right)+\left(a_{n-1}\left(n-1\right),n-1\right)+\ldots +\left(a_{1},1\right)\\ & =\left(\sum_{i=0}^{n}a_{i}i,{\left(\prod_{i=0}^{n}i^{a_{i}i}\right)}^{1/\sum_{i=0}^{n}a_{i}i}\right). \end{array}$$By Lemma 1, the first component (i.e., A(d)) is equal to |d(1)|; by the same lemma, we can also rewrite the second component (i.e., W(d)) as
$$W\left(d\right)={\left(\prod_{i=0}^{n}i^{a_{i}i}\right)}^{\frac{1}{A(d)}}$$
so it simply remains to justify why this is equal to ${\left|{\mathrm{Dir}}_{/d(0)}\left(d,d\right)\right|}^{\frac{1}{A(d)}}$. But a morphism in ${\mathrm{Dir}}_{/d(0)}\left(d,d\right)$ is exactly the data of an endomorphism of each fibre of $\pi_{d}:d\left(1\right)\to d\left(0\right)$; since there are $a_{i}$ fibres of size *i*, endomorphisms of these fibres are in bijection with the $a_{i}$-fold product of $i^{i}$, which is equal to $i^{a_{i}i}$, whence the claim. □

**Corollary** **3.**
*Let d∈Dir. Then the width W(d) is an algebraic number, i.e., the image of h:Dir→Rect lies in the sub-rig whose underlying set is $N\times {\bar{Q}}_{⩾0}$, where $\bar{Q}$ is the algebraic closure of Q.*


**Proof.** By Corollary 2, both $W{\left(d\right)}^{A(d)}$ and A(d) are equal to the cardinality of some sets, and thus integer. □

**Example** **5.**
*Reassuringly, if we start with a “rectangle”, then the area and width are exactly what we might expect. More concretely: consider $d\left(y\right)=a\cdot n^{y}$ for some a,n∈N; then, by Corollary 2,*
A(d)=d(1)=an,
*and, by direct calculation,*
W(d)=n.
*Comparing this to the picture of $a\cdot n^{y}$, we can explain why we chose the terminology “width” and “area”:*


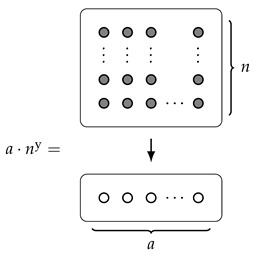


*Indeed, the area is exactly the number of dots in the (upper) rectangle, and the width is its width.*

*However, this picture now leads us to consider the question of whether or not there is a good meaning we can give to the “length” of this rectangle (which, here, should be equal to a). Indeed, this has been our motivation all along; we will return to this question in Example 9.*


**Example** **6.**
*The fact that d(1) is “rectangular” in Example 5 makes the terminology look like a numerical coincidence, but we can try to hone our intuition of what this really “means” by considering another example.*

*Let’s consider $d\left(y\right)=4^{y}+4$, which has area A(d)=d(1)=8. We can calculate its width by using the fact that*
$$\begin{array}{cc} h(4^{y}) & =(4,4)\\ h(4) & =h\left(1^{y}\right)+h\left(1^{y}\right)+h\left(1^{y}\right)+h\left(1^{y}\right)\\ & =(4,1) \end{array}$$
*whence*
$$\begin{array}{cc} h(d) & =(4,4)+(4,1)\\ & =\left(8,{\left(4^{4}1^{4}\right)}^{\frac{1}{8}}\right)\\ & =(8,2) \end{array}$$
*and so W(d)=2.*
*How, then, does the rectangle with area 8 and width 2 relate to our Dirichlet polynomial $d\left(y\right)=4^{y}+4$? That is, what is the process that takes us from d to $4\cdot 2^{y}$? Looking at the pictures of the bundles, we see that the width tells us how our bundle would look if we had the same set (d(1)) of draws, but with different outcomes, now all* equally likely*:*

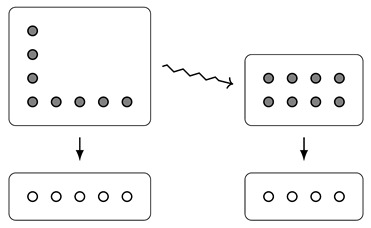


*Note that, in order to have equally sized fibres, we needed to have 4 outcomes, not 5 (since 8/2=4). We make this idea more precise (as well as explain why the rectangle is of size 4×2 instead of 2×4) in Section 6.*


**Example** **7.**
*We have just seen that $d\left(y\right)=4^{y}+4$ has W(d)=2 and A(d)=8, but now let’s look at an example where the numbers don’t divide so neatly.*

*Let $d\left(y\right)=4^{y}+3$. Then A(d)=d(1)=7, and, as in Example 6, we use the fact that*
$$\begin{array}{cc} h(d) & =(4,4)+(3,1)\\ & =\left(7,{\left(4^{4}1^{3}\right)}^{\frac{1}{7}}\right)\\ & =(7,2\sqrt[{7}]{2})\\ & \approx (7,2.21) \end{array}$$

*Of course, now we can’t draw a nice rectangle representing the evenly distributed bundle as we did in Example 6 for $4^{y}+4$, since we would have to have an outcome set of size 7/2.21≈3.17 elements, with fibres all of size 2.21, but this should come as no surprise, since 7 is prime. One might be tempted to solve this problem using groupoid cardinality (cf. [4]), but there are some technical issues here.*


## 5. Length

**Remark** **3.**
*We write log to mean $\log_{2}$.*


**Definition** **7.***Given a Dirichlet polynomial $d(y):=\sum_{i\in d(0)}d{\left[i\right]}^{y}$, we define its* entropy *H(d) by*
$$H(d):=-\sum_{i\in d(0)}\frac{\left|d[i]\right|}{\left|d(1)\right|}\log\left(\frac{\left|d[i]\right|}{\left|d(1)\right|}\right).$$*We then define its* length *(also called the* perplexity) *L(d) by*
$$L(d):=2^{H(d)}.$$

Readers might recognise H(d) as being the *Shannon entropy* of the corresponding probability distribution (cf. [5]). The convention for naming the sides of a rectangle is from [6] (Figure 1).

**Example** **8.**
*Consider $d\left(y\right)=n^{y}$ for some n∈N.*


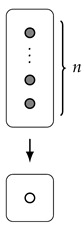


*Then d(0)=1 and d(1)=n, and so*
$$\begin{array}{cc} H(d) & =-\sum_{i\in \underline{1}}\frac{n}{n}\log\left(\frac{n}{n}\right)\\ & =-\log1\\ & =0 \end{array}$$
*whence $L\left(d\right)=2^{0}=1$.*

*In terms of distributions, this corresponds to the fact that the unique probability distribution on a single outcome has entropy equal to 0 (and so the same is true for any empirical distribution on a single outcome).*


**Example** **9.**
*Continuing on from Example 5, we can calculate the entropy of a uniform distribution on a many outcomes $d\left(y\right)=a\cdot n^{y}$ as*
$$\begin{array}{cc} H(a\cdot n^{y}) & =-\sum_{i\in \underline{a}}\frac{n}{an}\log\left(\frac{n}{an}\right)\\ & =-\log\left(\frac{1}{a}\right)\\ & =\log a \end{array}$$
*whence $L\left(a\cdot n^{y}\right)=2^{\log a}=a$, exactly as desired.*


**Example** **10.**
*Continuing on from Example 7, recall that $d\left(y\right)=4^{y}+4$ has area A(d)=8 and width W(d)=2. We can further calculate that*
$$\begin{array}{cc} H(d) & =-\sum_{i\in \underline{5}}\frac{\left|d[i]\right|}{8}\log\left(\frac{\left|d[i]\right|}{8}\right)\\ & =-\frac{4}{8}\log\left(\frac{4}{8}\right)-4\cdot \frac{1}{8}\log\left(\frac{1}{8}\right)\\ & =-\frac{1}{2}\log\left(\frac{1}{16}\right)\\ & =2 \end{array}$$
*whence $L\left(d\right)=2^{2}=4$.*

*As for $d\left(y\right)=4^{y}+3$, recall that its area is A(d)=7 and its width is $2\sqrt[{7}]{2}$. Now, its entropy is*
$$\begin{array}{cc} H(d) & =-\sum_{i\in \underline{4}}\frac{\left|d[i]\right|}{7}\log\left(\frac{\left|d[i]\right|}{7}\right)\\ & =-\frac{4}{7}\log\left(\frac{4}{7}\right)-3\cdot \frac{1}{7}\log\left(\frac{1}{7}\right)\\ & =\frac{\log7}{\log2}-\frac{8}{7} \end{array}$$
*and so its length is*
$$L\left(d\right)=2^{\frac{\log7}{\log2}-\frac{8}{7}}=\frac{7}{2\sqrt[{7}]{2}}.$$


Note that, in the above example, even though both L(d) and W(d) have non-integer values, the formula implied by our choice of nomenclature still holds: the area A(d) is equal to the length L(d) times the width W(d). This leads us to our main theorem. It says that the Shannon entropy, which is only homomorphic in products of distributions, can be computed in terms of the width and area, which together are homomorphic in both sums and products of distributions. We will explain this in more detail in Section 6.

**Theorem** **1.***For all d∈Dir, we have the* rectangle-area formula
A(d)=L(d)W(d).

**Proof.** In the following, we omit absolute value signs, writing e.g., d(1) instead of |d(1).First, write
$$d(y):=\sum_{j=0}^{n}a_{j}j^{y}≅\sum_{i\in d(0)}d{\left[i\right]}^{y}.$$
Now we can rewrite the length as
$$\begin{array}{cc} L(d) & =2^{H(d)}\\ & =2^{-\sum_{i\in d(0)}\frac{d[i]}{d(1)}\log\left(\frac{d[i]}{d(1)}\right)}\\ & =\prod_{i\in d(0)}2^{-\frac{d[i]}{d(1)}\log\left(\frac{d[i]}{d(1)}\right)}\\ & =\prod_{i\in d(0)}{\left(2^{-\log\left(\frac{d[i]}{d(1)}\right)}\right)}^{\frac{d[i]}{d(1)}}\\ & =\prod_{i\in d(0)}{\frac{d(1)}{d[i]}}^{\frac{d[i]}{d(1)}}\\ & =\frac{\prod_{i\in d(0)}d{\left(1\right)}^{\frac{d[i]}{d(1)}}}{\prod_{i\in d(0)}d{\left[i\right]}^{\frac{d[i]}{d(1)}}} \end{array}$$The numerator is then
$$\begin{array}{cc} \prod_{i\in d(0)}d{\left(1\right)}^{\frac{d[i]}{d(1)}} & =d{\left(1\right)}^{\sum_{i\in d(0)}\frac{d[i]}{d(1)}}\\ & =d(1)\\ & =A(d) \end{array}$$
since $\sum_{i\in d(0)}\left|d\left[i\right]\right|=\left|d\left(1\right)\right|$, by Corollary 1, and we can then apply Corollary 2. The denominator is exactly
$${\left(\prod_{i\in d(0)}d{\left[i\right]}^{d[i]}\right)}^{\frac{1}{d(1)}}$$
and so, by Corollary 2, we only need to justify why $\prod_{i\in d(0)}d{\left[i\right]}^{d[i]}$ is equal to $|{\mathrm{Dir}}_{/d(0)}\left(d,d\right)|$. However, this follows from the definition of an element of the latter set: a choice of map d[i]→d[i] for all i∈d(0). □

## 6. Interpreting Area, Length, and Width

We have mentioned many times that Dirichlet polynomials are equivalent to set-theoretic bundles, so the natural question to ask is “*why, then, should we work with the former instead of the latter?*”. One answer to this is question is the fact that *entropy does not respect bundle morphisms*: we cannot functorially assign a morphism between entropies to morphisms, since we are working with **arbitrary** morphisms of bundles. (If, however, we restrict to only morphisms given by pushforward, then [7] tells us (via *Faddeev’s theorem*) that the only possible functorial definition of entropy is given by the *relative entropy*, i.e., the difference of the entropies of the source and the target). This makes it seem rather bad to work with a *category* (such as that of bundles) instead of simply a *rig* (such as that of Dirichlet polynomials). Of course, this isn’t an entirely satisfactory answer, since we *do* care about the notion of morphisms for Dirichlet polynomials (for example, Corollary 2 tells us that the width can be expressed in terms of the number of certain morphisms). In light of Theorem 1, however, we might consider the following possibility: both area and length can be expressed in terms of d(0), d(1), and d[i] (for i∈d(0)), and we could *define* the width by W(d)A(d)/L(d).

A better answer to this question might be the following: the rig homomorphism h:Dir→Rect is incredibly simple, since it just maps $n^{y}$ to (n,n); from this computationally simple homomorphism, however, we can recover entropy (as log(A(d)/W(d))), without making any reference to the classical equation that defines it (“negative the sum of probabilities of the log of the probabilities”), but instead relying on the fact that Rect encodes the weighted geometric mean. That is, H(d) is only homomorphic in the product of distributions, whereas the pair (A(d),W(d)) is homomorphic in both the product and the sum.

Now, the entropy H(d)=logL(d) can be understood (via Huffman coding, cf. [8]) as *the average number of bits needed to code a single outcome* (over a long enough message). What is also true, however, is that the width (which is obtained purely “algebraically”, i.e., from the rig homomorphism h:Dir→Rect) gives similar information: by Theorem 1, combined with the previous sentence, logW(d) is *the average number of bits needed to code the draw, given an outcome* (in the same Huffman coding as before). This answers the question of “*what is special about the bundle defined by the width and length*” with “*it describes the optimal encoding of draws, given outcomes*”.

As for the picture in Example 6, we can now understand the hand-wavy explanation a bit better (but still just as hand-wavy-ly): we take our original “half-filled” rectangle d(1) and pour its contents into a new rectangle, of length L(d), and then “slosh the contents around” until they lie flat, and then put a lid on it; the rectangle will be perfectly filled up, and the placement of the lid will be given by W(d).

We also mentioned, in Corollary 3, that the width W(d) of any Dirichlet polynomial *d* is an algebraic number, but the actual result is slightly more interesting that this: Corollary 2 tells us that $W{\left(d\right)}^{A(d)}$ is equal to the cardinality of the set ${\mathrm{Dir}}_{/d(0)}\left(d,d\right)$. We already know how to understand endomorphisms of *d* that fix d(0) as endomorphisms of d(1) that fix the outcome; we can understand $W{\left(d\right)}^{A(d)}$ as maps from A(d) to W(d); roughly speaking, such a map f:A(d)→W(d) determines the remaining ambiguity in determining a draw, given its outcome.

## 7. Cross Entropy

Everything above can be viewed as a specific example of the analogous *cross* notions. That is, given two Dirichlet polynomials, we can define their cross area, cross width, etc. as follows.

**Definition** **8.***Let d,e∈Dir be Dirichlet polynomials such that d(0)=e(0). Then we define the*cross entropy *H(d,e) by*$$H\left(d,e\right)=-\sum_{i\in d(0)}\frac{\left|d[i]\right|}{\left|d(1)\right|}\log\left(\frac{\left|e[i]\right|}{\left|e(1)\right|}\right)$$*and the* cross area, cross width, *and* cross length *by*
$$\begin{array}{cc} A(d,e):= & \left|e(1)\right|\\ W(d,e):= & {\left|{\mathrm{Dir}}_{/d(0)}\left(d,e\right)\right|}^{\frac{1}{\left|d(1)\right|}}\\ L(d,e):= & 2^{H(d,e)} \end{array}$$
*(respectively).*

By definition, X(d,d)=X(d) for X∈{A,W,H,L}. That is, just as cross entropy is a generalisation of entropy, the notions of cross width, etc., generalise the notions of width, etc.

**Remark** **4.***Note that we can recover the notion of* relative entropy (*also known as* Kullback–Leibler divergence) *$D_{\mathrm{KL}}\left(p∥q\right)$, as studied in [9], from cross entropy:*
$$H\left(d,e\right)=H\left(d\right)+D_{\mathrm{KL}}\left(p∥q\right)$$
*(which can also be seen to justify the fact that H(d,d)=H(d)).*

**Remark** **5.**
*Although we have some idea of how to understand these cross notions (e.g., cross area can be understood as the number of “actual” draws, when we think of d as being a potentially inaccurate model for e), the choice of definitions in Definition 8 was chosen simply so that*
*1.* 
*we recover the “uncrossed” notions when we take d=e, and*
*2.* 
*Theorem 2 holds.*



**Theorem** **2.***For all d,e∈Dir, we have the* cross rectangle-area formula
A(d,e)=L(d,e)W(d,e).

**Proof.** This proof follows exactly the same argument as the proof of Theorem 1. □

## Figures and Tables

**Figure 1 entropy-23-01085-f001:**
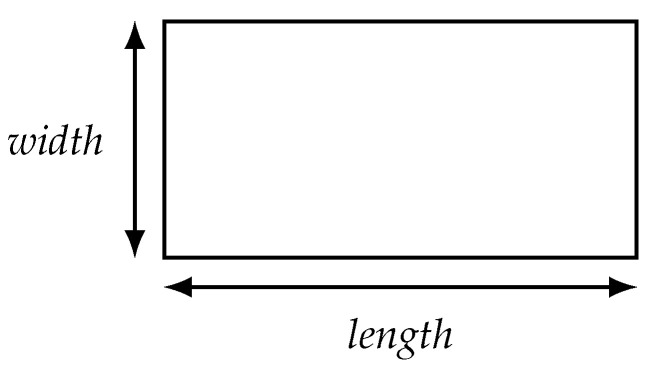
Our convention for naming the sides of a rectangle, from [6].

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
