# Peer review of "Dirichlet Polynomials and Entropy"

_entropy, 2021, doi:10.3390/e23081085_

Round 1

Reviewer 1 Report

The paper is well written and it is clear for the reader, but I suggest to add a paragraph where the conclusions are better summarized and where the further development and application of the study are outlined. 

Author Response

Thank you for your review. We have slightly edited §6 so that the conclusions of the paper are hopefully clearer.

Reviewer 2 Report

Accept in present form

Author Response

Thank you very much for your review.

Reviewer 3 Report

The authors study a characterization of the entropy of an empirical distribution based on a rig homomorphism related to Dirichlet polynomials. The entropy is calculated as the logarithm of the ratio of the area to the width of a Dirichlet polynomial corresponding to a given empirical distribution. In terms of area and width, the homomorphism preserves both the product and the sum in contrast to the entropy which preserves only the product. The results are generalized to the cross entropy. 

This paper provides an interesting category theoretic perspective on entropy different from [BF14]. It is well-organized and is comfortable to read. The reviewer recommends accepting the paper after minor revision taking into account the following comments.

1. The authors implicitly assume that the reader of the paper has a working knowledge of basic category theory. However, I am afraid that many readers of Entropy may not meet this requirement. Please indicate the necessary level of category theory and cite references. 

2. The authors briefly discuss the motivation for the generalization to the cross entropy in Remark 7.4 and prove the generalized version of the rectangle-area formula in Theorem 7.5. How about the results in Section 4? For example, can A(d,e) and W(d,e) be defined through a rig homomorphism? Is there a generalization of the second equation in Corollary 4.5? 

3. [BF14], [BFL11] and [BHW10] are published in journals. Please cite the published versions.

Author Response

Thank you for your constructive feedback.

  1. We have added a small section on categorical prerequisites, along with a link to an introductory textbook (freely available online).
  2. The second equation in Corollary 4.5 is actually how we define the notion of cross width (in Definition 7.1). We do not have any further results on the cross notions, and really include them here simply for completeness, because they follow almost immediately.
  3. Thank you for pointing this out. This has been fixed.